# Choroidal Volume Evaluation after Photodynamic Therapy Using New Optical Coherence Tomography Imaging Algorithm

**DOI:** 10.3390/ph14111140

**Published:** 2021-11-10

**Authors:** Miki Sato-Akushichi, Shinji Ono, Gerd Klose, Youngseok Song

**Affiliations:** 1Department of Ophthalmology, Asahikawa Medical University, Asahikawa 078-8510, Japan; o-shinji@asahikawa-med.ac.jp (S.O.); ysong@asahikawa-med.ac.jp (Y.S.); 2Carl Zeiss Meditec, Inc., Dublin, CA 94568, USA; gerd.klose@zeiss.com

**Keywords:** central serous chorioretinopathy, photodynamic therapy, choroidal volume, choroidal thickness, new algorithm

## Abstract

To evaluate choroidal volume and thickness changes after photodynamic therapy (PDT) for chronic central serous chorioretinopathy (CSC). Chronic CSC eyes with a history of PDT were selected. Average choroidal volume, average choroidal thickness, the maximum and minimum choroidal thickness of the macula irradiated area and peripheral non-irradiated areas before and after one and three months of treatment were examined. A total of 14 patients with chronic CSC and 9 controls without any eye pathology were enrolled. The mean choroidal volume in CSC before and, and after one and three months of treatment were 2.36 (standard deviation: 0.70), 1.90 (0.69), 1.86 (0.66) mm^3^ for the central area, 1.25 (0.38), 1.14 (0.35), 1.13 (0.34) mm^3^ for superior nasal area, 1.47 (0.41), 1.28 (0.43), 1.26 (0.43) mm^3^ for superior temporal area, 1.07 (0.49), 0.95 (0.38), 0.93 (0.35) mm^3^ for inferior nasal area, 1.17 (0.38), 1.04 (0.32), 1.03 (0.33) mm^3^ for inferior temporal area. This study revealed the choroidal volume changes in a short period after PDT and a decrease in unirradiated choroidal volume was also shown after the treatment. The algorithm provided on the ARI Network enables to evaluate the choroidal changes quantitatively and qualitatively.

## 1. Introduction

After the treatment of age-related macular degeneration with photodynamic therapy (TAP) study and the verteporfin in photodynamic therapy (VIP) study for age-related macular degeneration, an expanded indication of photodynamic therapy (PDT) has allowed to treat central serous chorioretinopathy (CSC) [1,2]. Standard PDT with verteporfin is an effective and safe treatment for chronic CSC with a significant improvement in the 4-year post-op period, both anatomically and visually [3]. However, choriocapillaris ischemia, secondary choroidal neovascularization, transient impairment in retinal function, and retinal pigment epithelium (RPE) atrophy are common side effects of PDT treatment [4,5,6]. To minimize the side effects, a reduced dose, fluence, and exposure time of PDT have been evaluated in CSC treatment. Reduced fluence PDT is effective for symptomatic subfoveal serous pigment epithelial detachments with hyper fluorescence on late-phase indocyanine green angiography (ICGA) [7]. A half dose of verteporfin is as effective as the conventional full dose PDT [8]. Half fluence and half dose PDT cause less ischemia at the level of the RPE/choriocapillaris [9]. Therefore, half dose or fluence PDT is the preferable standard treatment for chronic CSC [10].

The PDT causes choriocapillaris narrowing, choroidal hypoperfusion, reduction of choroidal exudation, and choroidal vascular remodeling, which includes normalization in calibers of the dilated and congested choroidal vasculature [4,11,12]. Structural damage after PDT was found in the treatment area [11], and most reports shed light on the irradiated choroidal area changes. PDT displays a vaso-occlusive mechanism, which affects even a healthy choroid, as seen in ICGA [13]. Additionally, few reports studied changes in peripheral nonirradiated choroidal area [14].

Most of the previous reports measuring choroidal thickness involved manual or semi-automated methods [15,16,17,18]. Choroid thickness measurements are complicated because the chorioscleral interface is a transient zone, and the definition of the posterior boundary remains inconstant [19]. In addition, it is well known that CSC has a thick choroid with related signal attenuation; choroidal thickness measurements between-examiner and within-examiner agreements are worse in a thick choroid than in a thin choroid [18].

Optical coherence tomography (OCT) is a technique that provides morphologic and functional tissue information, noninvasively in the short time. Further, Swept-source (SS) OCT technology enables higher scanning speeds, extensive depth range with significantly reduced sensitivity roll-off, reduced fringe washout from sample motion or rapid transverse scanning, reduced effect of distortion, and improved light detection efficacy due to dual balance detection. [20,21,22] Thus, SS-OCT enables visualization of the deep ocular tissues, such as the choroid or the sclera with less signal attenuation, penetrating through the RPE and Bruch’s membrane.

The Advanced Retina Imaging (ARI) network is a web-based platform to support clinical research with PLEX Elite 9000 SS-OCT and OCT angiography data. Among other algorithms, ZEISS provides a Choroid Quantification analysis, which allows to automatically visualize and quantify choroidal properties, such as choroidal volume and thickness, not only of the irradiated area but also of peripheral nonirradiated areas treated with PDT in chronic CSC. The purpose of this present study is to evaluate choroidal morphologic changes both in irradiated and peripheral non-irradiated areas treated with PDT in CSC.

## 2. Results

### 2.1. Patient Population

A total of 14 patients diagnosed chronic CSC (all patients were men) and 9 controls (all patients were also men) were enrolled. The mean age of the subject was 59.4 (15.85) years with CSC and 70.4 (7.55) with control. Mean refractive error was −0.29 (2.14, range: −4.25 to +2.75 diopters) with CSC and 1.22 (1.23, range: −0.25 to +2.75 diopters) for the control group. Mean spot size was 4521 μm (1201, range: 2000 to 6000 μm). Treatment conditions were five patients with half dose PDT, seven patients with half fluence PDT, and two patients with half time PDT. The demographics of patients and control are shown in Table 1.

### 2.2. Inter-Test Repeatability

Firstly, inter-test repeatability was confirmed using seven participants data, randomly selected. All grid areas (central, inner nasal, inner superior, inner temporal, inner inferior, outer nasal, outer superior, outer temporal, outer inferior, extended C1 nasal superior, extended C1 nasal inferior, extended C1 temporal superior, extended C1 temporal inferior, extended C1 superior nasal, extended C1 inferior nasal, extended C1 superior temporal, extended C1 inferior temporal, extended C2 nasal superior, extended C2 nasal inferior, extended C2 temporal superior, extended C2 temporal inferior) were applied for measurement. Inter-test repeatability was very high, with an intraclass correlation coefficient (ICC) of 0.961 (95% confidence interval (CI), 0.946–0.971) for choroidal volume, 0.955 (95% CI, 0.939–0.968) for average choroidal thickness, 0.881 (95% CI, 0.839–0.912) for maximum choroidal thickness, and 0.944 (95% CI, 0.923–0.959) for minimum choroidal thickness.

### 2.3. Changes of Choroidal Volume

Choroidal volume between patients before and after the PDT and control are shown in Figure 1 and Appendix A. Eyes with chronic CSC exhibited thickened choroidal volume in the central area. The post treatment choroidal volume showed significant reduction from baseline at all time points after the PDT, including both irradiated central area and non-irradiated peripheral areas; although such changes were not found in the inferior nasal area three months after the PDT. Choroidal volume decreased by 19.5% in the central area and 8.8–12.9% in the peripheral areas after one month of treatment and decreased by 21.2% in the central area and 9.6–14.3% in the peripheral areas after three months of treatment. There were no significant differences from one month to three months after the PDT in all areas (*p* value > 0.05).

### 2.4. Chenges of Choroildal Thickness

Average choroidal thickness and maximum and minimum choroidal thickness between patients before and after the PDT and control are shown in Figure 2, Figure 3 and Figure 4 and Appendix A. Among average choroidal thickness, the eyes with chronic CSC had thicker choroid in the central area. After the treatment, the mean choroidal thickness showed a significant reduction from baseline at all time points after the PDT including both irradiated central area and non-irradiated peripheral areas except inferior nasal area three month after the PDT. For maximum choroidal thickness, there were no statistically significant differences in all areas between chronic CSC and control. (*p* value > 0.05). The post treatment maximum choroidal thickness showed significant reduction from the baseline at all time points after the PDT in all areas, both one and three months after treatment. For minimum choroidal thickness, eyes with chronic CSC exhibited thicker choroid in the central area. The post treatment minimum choroidal thickness showed significant reduction from baseline in central area and superior temporal and inferior temporal areas one month after the PDT. There was no significant difference in inferior nasal area after one month and three months after the PDT compared to pretreatment.

### 2.5. Case Presentations

#### 2.5.1. Case 1: A 46 Year-Old-Male

A 46-year-old male complained of metamorphopsia for three years in his left eye. Logarithm of the minimum angle of resolution (LogMAR) in his left eye was −0.0792. The subretinal detachment and the RPE abnormality were seen at the macular area in OCT. Fluorescein angiography (FA) showed multiple leak points and ICGA showed dilated choroidal vessels and choroidal vascular hyperpermeability. After applied half dose PDT (spot size = 4000 μm), the subretinal detachment disappeared completely until one year of follow up. The mean choroidal volume before treatment, one month after treatment, and three months after treatment was 3.00, 2.47, 2.72 mm^3^ for the central area, 1.77, 1.46, 1.62 mm^3^ for superior nasal area, 2.05, 1.70, 1.90 mm^3^ for superior temporal area, 1.48, 1.28, 1.42 mm^3^ for inferior nasal area, 1.88, 1.61, 1.77 mm^3^ for inferior temporal area. Visual acuity did not change through the following period (Figure 5).

#### 2.5.2. Case 2: A 53-Year-Old Male

A 53-year-old male was referred to our hospital, with a finding of subretinal detachment of more than one year in his right eye. LogMAR in his right eye was 0.22, and OCT revealed one disc diameter sized shallow subretinal fluid and thick choroid with inner choroidal attenuation. FA showed a diffuse leakage and ICGA showed dilated choroidal vessels and choroidal vascular hyperpermeability. Half dose PDT (spot size = 5400 μm) was effective. The mean choroidal volume before treatment, one month after treatment, and three months after treatment was 2.46, 1.85, and 2.00 mm^3^ for the central area, 1.47, 1.41, and 1.40 mm^3^ for the superior nasal area, 1.61, 1.48, and 1.48 mm^3^ for the superior temporal area, 1.12, 1.08, and 1.08 mm^3^ for the inferior nasal area, 1.28, 1.19, and 1.21 mm^3^ for the inferior temporal area. Visual acuity did not improve because of retinal outer layer atrophy (Figure 6).

## 3. Discussion

In this study, we focused on the choroidal changes in both PDT irradiated area and peripheral non irradiated areas two-dimensionally and three-dimensionally using a new developed algorithm. We discovered that PDT treatment affects choroidal volume and thickness not only in the irradiated area but also in the non-irradiated areas.

Previous reports conducted choroidal thickness measurements using manual or semi-automated methods [15,16,17,18]. However, measuring choroid thickness is not easy because the chorioscleral interface is a transient zone, and the definition of a posterior boundary remains inconstant [19]. In addition, the manual measurement is time consuming and requires training to scale accurately [18,23]. Moreover, due to signal attenuation, choroidal thickness measurements between-examiner and within-examiner agreements are worse in a thick choroid than in a thin choroid [18]. This study applied the algorithm provided on the ARI network to automatically measure choroidal volume, average choroidal thickness, and maximum and minimum choroidal thickness. We first confirmed inter-test repeatability including all of the areas overlaid in the image, then conceded high ICC in each parameter. This algorithm automatically and accurately measured the choroidal volume and thickness.

Subfoveal choroidal thickness decreased after PDT for CSC [9,14,24]. Post full-fluence and half-fluence PDT subfoveal choroidal thickness values were significantly reduced at 3, 6, and 12 months compared with the baseline, and the mean extent of reduction is significantly greater in the full-fluence PDT than half-fluence PDT [24]. In addition, by applying reduced-fluence PDT, the reduction of choroidal thickness was statistically significantly different between pretreatment and one month after treatment, and pretreatment and three months after treatment, even though there were no statistically significant differences between one month after treatment and three months after treatment [14]. Our result exhibited the same result for the choroidal volume and average choroidal thickness in the irradiated central area. Even in non-irradiated peripheral areas, except for the inferior nasal area, there was a reduction of choroidal volume and thickness. Divided into maximum and minimum choroidal thickness, all areas, irradiated and non-irradiated, showed significantly reduced maximum choroidal thickness at one month and three months after treatment, whereas central, peripheral superior, and peripheral temporal areas showed significantly reduced minimum choroidal thickness at three months after treatment. It is reported that the superior quadrant exhibited the highest choroidal volume while the nasal quadrant had the lowest choroidal volume in healthy subjects [25]. Additionally, men have a significantly greater choroidal volume than women [25], therefore it seems that there were no significant differences between CSC eyes and control in the choroidal volume and thickness in the peripheral areas. In addition, this infers that the inferior nasal area tends to be a thin choroid, and the effect of treatment on the normalization in the choroid may be small.

The choroid arteries arise from the long and short posterior ciliary arteries and branches of Circle of Zinn, draining to vortex veins that merge with the ophthalmic vein. Among the three vascular layers of choroid, Haller’s layer includes large arteries and veins, and Sattler’s layer is composed of medium and small arterioles that feed the capillary network of the choriocapillaris and venules [26]. The choriocapillaris is a highly anastomosed network of capillaries, and the choroidal micro vessels are fenestrated [27]. A total of 50% of non-pathogenic individuals show asymmetry of choroidal venous drainage [28] and among them, two thirds of the eyes have preferential direction to superotemporal route and the other to inferotemporal or superonasal route [28]. Superior and inferior vortex vein anastomosis is prominent in CSC [29]. Moreover, in the patients with pachychoroid related conditions, such as CSC, the anastomotic connections were found among the superonasal, superotemporal, and inferotemporal vortex vein system [30]. The mechanism of PDT in CSC is presumably based on the formation of free radicals upon illuminated treatment area, which leads to vascular endothelium damaging, choriocapillaris narrowing, hypoperfusion and subsequent choroidal vascular remodeling [4,10,11,12]. The mean diameter of the dilated choroidal vessel decreases after the PDT [4], and PDT reduces the vascular density in the choriocapillaris and middle layer of the choroid [31]. This study revealed that the choroidal volume and thickness decreased even in non-irradiated area of PDT eyes. It is presumed that PDT mainly affects the irradiated area and induces decrease of blood supply, which changes to choroidal vessel condition. This leads peripheral choroidal volume and thickness to decrease, due to anastomosis, asymmetry of watershed zone, high permeability, and upstream vessels, which supply the macula area.

There are several limitations to this study. First, as a retrospective study, a patient selection bias could have played a role. Second, despite the very small sample size, statistically significant findings were seen both in the irradiated area and non-irradiated areas after the PDT. Thus, large scale, randomized, and prospective studies are warranted. Also, in spite of the very small sample size, there was more than one treatment condition. Although we conducted separate analyses of three treatment groups, however, it would be desirable to conduct large scale samples. (Appendix A). Third, we could not remove the confounding factor of diurnal variations of the choroidal thickness due to images not being obtained within a limited period of time. Additionally, aging [25,32] and axial length affect the choroidal volume [25]. Unable to perform A-scan ultrasound, we used refractive errors to reduce the variability. Lastly, we did not analyze choroid stromal and luminal area separately. There is a correlation between choroidal vessels and thickness [33,34]. A half dose of PDT produced short-term changes on the luminal component of both choriocapillaris and choroid [35], hence it seems that the choroidal vessels shrank, and the volume of stroma did not change after the PDT. However, we did not develop a precise algorithm to analyze the choroidal luminal and stromal area. Thus, we did not disclose the relationships between choroidal vessels and stroma in this study.

## 4. Materials and Methods

### 4.1. Population

Patients with diagnosis of chronic CSC at Asahikawa Medical University Hospital from September 2019 to March 2021 were enrolled. Chronic CSC diagnosis was based on clinical characteristics of multimodal imaging lasting 6 months or more or recurrence. Patients involved in this study underwent comprehensive eye examinations including best-corrected visual acuity (BCVA), slit-lamp biomicroscopy for anterior and posterior segment examination, fundus photography, FA and ICGA, OCT (Spectralis HRA + OCT; Heidelberg Engineering, Heidelberg, Germany), and OCT angiography (PLEX Elite 9000; Carl Zeiss Meditec, Dublin, CA, USA). BCVA was converted to logMAR and used for statistical analysis. OCT scans and OCT angiography images were obtained at baseline, one month, and three months after the PDT. The inclusion criteria were (1) age > 40 years, (2) documented subretinal fluid in the foveal region with or without serous retinal RPE detachment on OCT image, (3) macular RPE decompensation with subtle or diffuse leaks with RPE detachment on FA, (4) abnormal, dilated choroidal vascular hyperpermeability on ICGA, and (5) absence of choroidal neovascularization. Exclusion criteria were (1) high refractive error (less than −6 diopters or more than +3 diopters), (2) the presence of other retinal diseases including diabetic retinopathy, retinal vein occlusion, age related macular degeneration, (3) any history of intraocular surgery within the past 6 months, (4) any history receiving anti-vascular endothelial growth factor (VEGF) injections before PDT, (5) low quality OCT angiography images because of media opacities, poor cooperation with taking images.

For control individuals without any eye pathology, the written informed consent was obtained before administering a comprehensive eye examination including BCVA, slit-lamp biomicroscopy for anterior and posterior segment examination, fundus photography, and OCT/OCT angiography. Exclusion criteria were identical to subjects with chronic CSC.

### 4.2. Treatment Conditions

Intravenous verteporfin was administered over a period of 10 min. A diode laser emitting at 689 nm coupled into a slit lamp system (Coherent Inc., Palo Alto, CA, USA) was used at an irradiance of 600 mW/cm^2^ over 83 s, starting 15 min after the start of the infusion. The treatment spot was determined by using the diameter of the largest circle that covered the area of leakage by FA or the area of hyperpermeability by ICGA or both. An additional 1000 μm margin was added to ensure a complete coverage of the lesion during light exposure. Choice of half of the normal dose, half of the normal fluence, or half of the normal time PDT was determined by the treating provider (S.O.). The standard light energy was 50 J/cm^2^, verteporfin dose was 6 mg/m^2^, and irradiation time was 83 s, and therefore the total light energy was set at 25 J/cm^2^ for the half fluence approach, the total verteporfin dose was set at 3 mg/m^2^ body for the half dose approach, and the total irradiation time was set at 42 s for the half time approach. Patients were advised to avoid direct sunlight exposure for three days after treatment.

### 4.3. Image Acquisition and Analysis

OCT angiography images were taken on the PLEX Elite 9000 SS-OCT, which operates at 1060 nm central wavelength, 100,000 A-scans per second with axial resolution of 6.3 μm, and transverse resolution of 20 μm in tissue. A-scan depth penetration in tissue was 3.0 mm and 12 mm × 12 mm OCT angiography images were used for the analysis. The scans were analyzed on the ARI Network platform, which provides a fully automated Choroid Quantification algorithm developed by ZEISS. Figure 7 shows a choroidal volume map obtained after automatic segmentation of the choroidal edge. Color maps display a range of 0 to 800 microns. Referring to the early treatment diabetic retinopathy study (ETDRS) grid, five circles were overlayed onto the image; center fovea (1 mm diameter), inner ring (0.5–1.5 mm from the center), outer ring (1.5–3 mm from the center), extended C1 ring (3–4.5 mm from the center), and the extended C2 ring (4.5–6 mm from the center). The inner ring and outer ring were divided into four parts: superior, inferior, nasal, temporal, and the extended C1 ring was divided into eight parts: superior nasal, superior temporal, inferior nasal, inferior temporal, nasal superior, nasal inferior, temporal superior, and temporal inferior. The extended C2 circle was divided into four parts (nasal superior, nasal inferior, temporal superior, and temporal inferior) but superior and inferior parts were not applied. The optic disc area (colored in black) was excluded from the evaluation. The applied areas were determined in the following five parts; central area (3000 μm in diameter, merged center fovea and inner ring) and superior nasal, superior temporal, inferior nasal and inferior temporal area, 1500 μm concentric apart from the center circle, named as the extended C1 area before, respectively (Figure 8). Average choroid volume, choroidal thickness, and maximum and minimum choroidal thickness of macula irradiated area and peripheral non-irradiated area before and after one and three months of treatment were examined.

### 4.4. Statistical Analysis

Statistical analysis was performed by the use of SPSS version 25, free software R version 4.0.5 (The R Foundation for Statistical Computing Platform, Vienna, Australia) and EZR (Jichi Medical University, Saitama, Japan), a graphical user interface for R [36]. The Bonferroni methods’ multiple comparison was used for changes in choroidal volume and thickness in each sector after the PDT, and the Mann-Whitney *U* test was used for the comparison between PDT treated eyes baseline and control eyes. In addition, the Steel-Dwass test was used for multiple comparisons in each area between different treatment conditions. Data are presented by mean (standard deviation), and a *p*-value of <0.05 was considered as statistically significant.

## 5. Conclusions

This study revealed choroidal morphological changes including choroidal volume and thickness, in the short-term post-treatment. The choroidal volume and choroidal thickness decreased even in non-PDT irradiated area after the treatment, and among the peripheral non irradiated area, the superior area was significantly affected, in addition to the irradiated central area. This new algorithm can assist in the evaluation of choroidal changes quantitatively and qualitatively.

## Figures and Tables

**Figure 1 pharmaceuticals-14-01140-f001:**
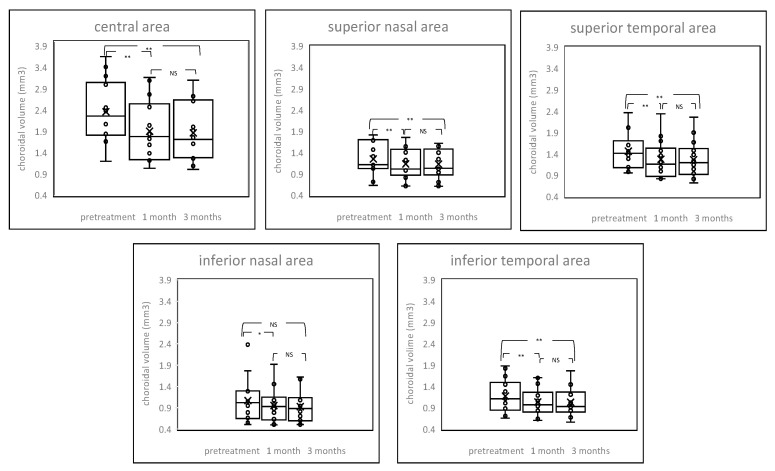
Box-plot graphs of choroidal volume before and after treatment. The post treatment choroidal volume showed a significant reduction from baseline at all time points after the PDT, including both irradiated central area and non-irradiated peripheral areas. NS = not significant, * *p* < 0.05, ** *p* < 0.01.

**Figure 2 pharmaceuticals-14-01140-f002:**
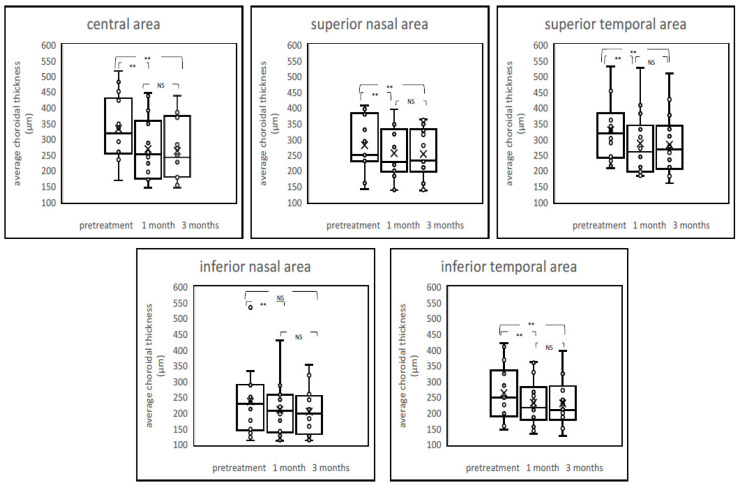
Boxplot graphs of average choroidal thickness before and after treatment. The post-treatment average choroidal thickness showed a significant reduction from baseline at all time points after the PDT, including both the irradiated central area and nonirradiated peripheral areas. NS = not significant, ** *p* < 0.01.

**Figure 3 pharmaceuticals-14-01140-f003:**
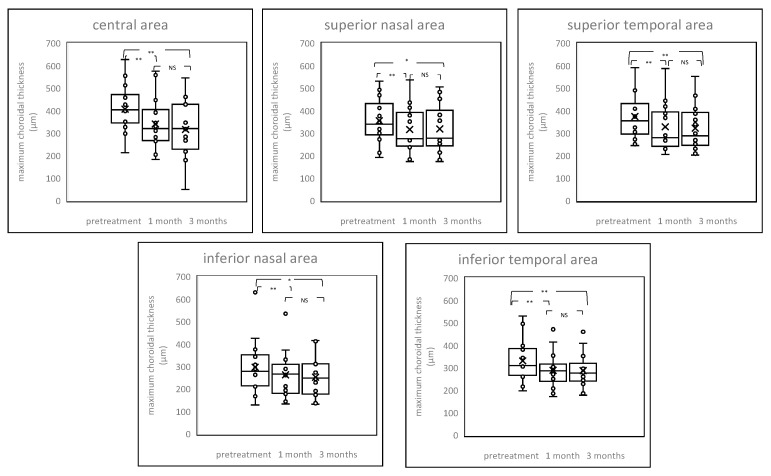
Box-plot graphs of maximum choroidal thickness before and after treatment. The post treatment maximum choroidal thickness showed significant reduction from baseline at all time points after the PDT in all areas, both one and three months after treatment. NS = not significant, * *p* < 0.05, ** *p* < 0.01.

**Figure 4 pharmaceuticals-14-01140-f004:**
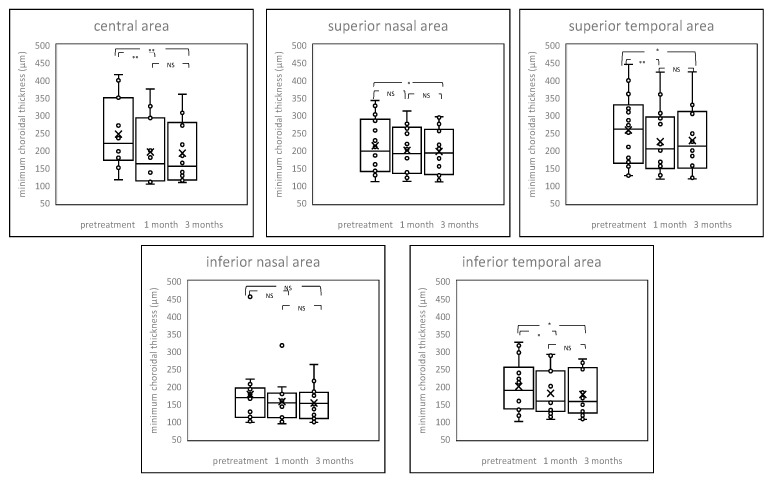
Box-plot graphs of minimum choroidal thickness before and after treatment. The post treatment minimum choroidal thickness showed significant reduction from baseline at central area, superior temporal, and inferior temporal areas one month after the PDT. NS = not significant, * *p* < 0.05, ** *p* < 0.01.

**Figure 5 pharmaceuticals-14-01140-f005:**
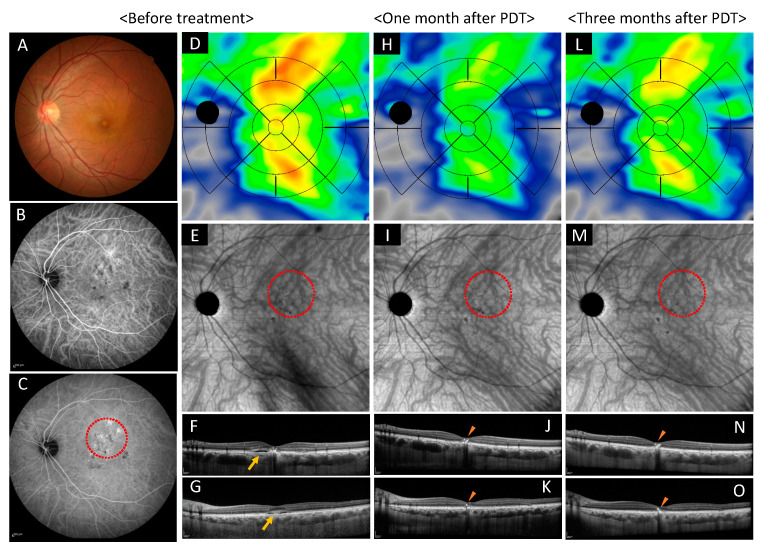
Images of the left eye of a 46-year-old male with chronic central serous chorioretinopathy. (**A**–**G**) Show before treatment, (**H**–**K**) show one month after the photodynamic therapy (PDT), and (**L**–**O**) show three months after the PDT. (**A**) Color fundus photograph shows the subretinal detachment and the retinal pigment epithelium abnormality at the macular area. (**B**,**C**) Early and late phase of indocyanine green angiography show dilated choroidal vessels, punctate hyperfluorescent spots, and choroidal vascular hyperpermeability. PDT irradiated the area highlighted with red dot circle. (**D**) Choroidal volume color map shows thickened choroid at superior, macular, and inferior areas before treatment. (**H**) Choroidal volume decreased at the superior, macular, and inferior areas after one month. (**L**) Three months after treatment, choroidal volume thickened again at the superior and inferior areas. (**E**) En face optical coherence tomography images show dilated choroidal vessels and lost watershed zone accompanied by superior and inferior choroidal vessel anastomosis before treatment. (**I**) Constriction of choroidal blood vessels was observed one month after treatment. (**M**) Three months after treatment, constriction of choroidal blood vessels was still observed. (**F**,**J**,**N**) are vertical scans and (**G**,**K**,**O)** are horizontal sans through central fovea. (**F**,**G**) show shallow subretinal detachment before treatment (show arrow). (**J**,**K**) Serous retinal detachment resolved completely at one month after treatment, however intraretinal hyperreflective materials remained (show arrowhead). (**N**,**O**) No recurrence three months after treatment. Intraretinal hyperreflective materials remained (see arrowheads).

**Figure 6 pharmaceuticals-14-01140-f006:**
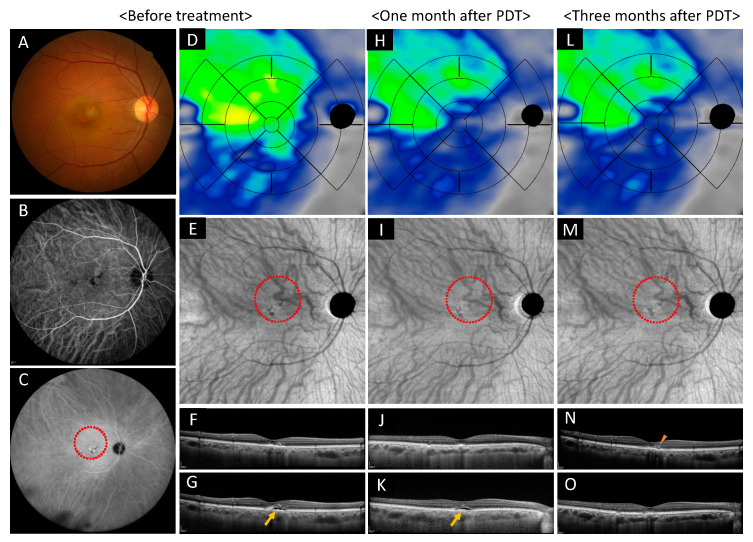
Images of the right eye of a 53-year-old male with chronic central serous chorioretinopathy. (**A**–**G**) Show before treatment, (**H**–**K**) show one month after photodynamic therapy (PDT), and (**L**–**O**) show three months after PDT. (**A**) Color fundus photograph shows the subretinal detachment and the retinal pigment epithelium (RPE) abnormality at the macular area. (**B**,**C**) Early and late phase of indocyanine green angiography show dilated choroidal vessels, punctate hyperfluorescent spots, and choroidal vascular hyperpermeability. PDT irradiated the area highlighted with red dot circle. (**D**) Choroidal volume color map shows thickened choroid at superior and temporal areas before treatment. (**H**) The choroidal volume decreased at superior, macular, and temporal areas at one month. (**L**) Three months after treatment, choroidal volume showed no change at the superior, macular, and temporal areas. (**E**) En face optical coherence tomography images show dilated choroidal vessels and dilated abnormal anastomosed vessels crossed superior and inferior area. (**I**) Constriction of the choroidal blood vessels was observed one month after treatment. (**M**) Three months after treatment, constriction of choroidal blood vessels was still observed. (**F**,**J**,**N**) are vertical scans and (**G**,**K**,**O**) are horizontal scans through central fovea. (**F**,**G**) Show shallow subretinal detachment before treatment (show arrow). (**J**,**K**) Shallow retinal detachment still exists one month after the treatment (show arrow). (**N**,**O**) Serous retinal detachment resolved completely at three months after the treatment, remained RPE undulation and intraretinal hyperreflective materials (show arrowhead).

**Figure 7 pharmaceuticals-14-01140-f007:**
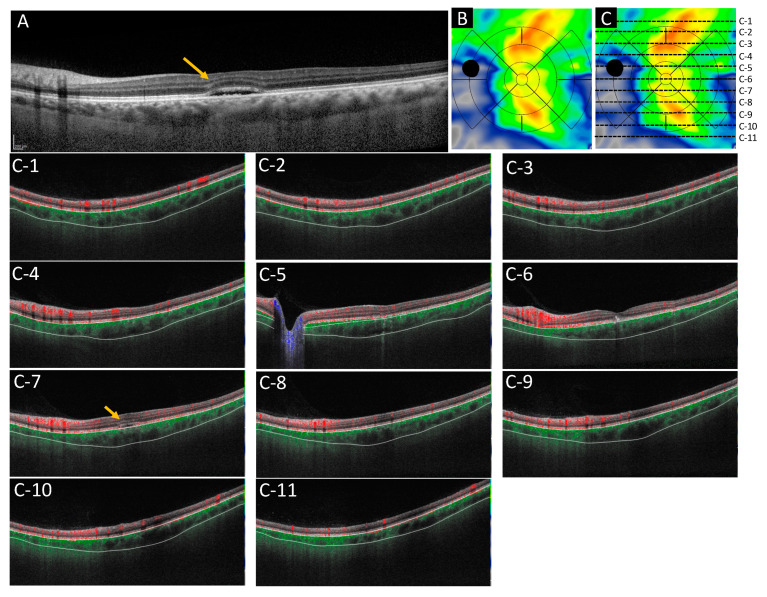
Images of choroidal volume map. (**A**) Shows the optical coherence tomography horizontal scan diagnosed chronic serous chorioretinopathy. (**B**) Shows the choroidal volume map in this case. (**C**) Shows each localization in which measurements were done. (**C-1**–**C-11**) Show the choroidal-scleral interface by scan from top to bottom in the image, corresponding to each line drown in (**C**). It is segmented correctly. The color map display ranges from 0 to 800 microns. The arrows show the subretinal fluid.

**Figure 8 pharmaceuticals-14-01140-f008:**
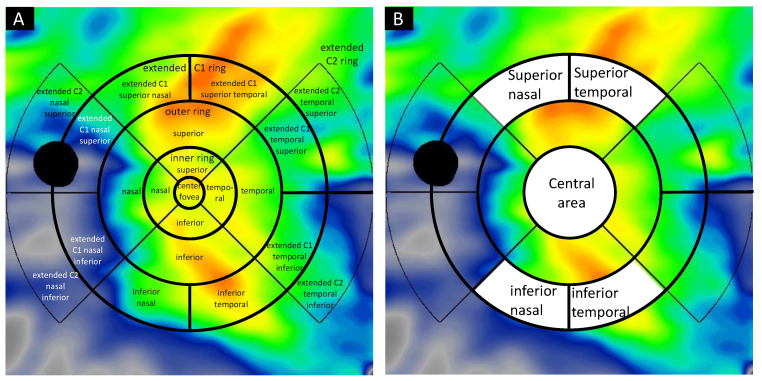
Choroidal volume map overlayed the extended ETDRS grid. (**A**) Images of choroidal map overlaid the extended early treatment diabetic retinopathy study (ETDRS) grid, which shows the center fovea (1 mm diameter), inner ring (0.5–1.5 mm from the center), outer ring (1.5–3 mm from the center), extended C1 ring (3–4.5 mm from the center), extended C2 ring (4.5–6 mm from the center). The inner ring and outer ring were divided into four parts: superior, inferior, nasal, and temporal, and the extended C1 ring was divided into eight parts: superior nasal, superior temporal, inferior nasal, inferior temporal, nasal superior, nasal inferior, temporal superior, and temporal inferior. The extended C2 ring was divided into four parts: nasal superior, nasal inferior, temporal superior, and temporal inferior, but the superior and inferior parts were not applied. The optic disc was colored as a black and ruled out from the calculation. (**B**) Areas applied in this study were determined in the following five parts; the central area (3000 μm in diameter, merged center fovea and inner ring) and the extended C1 superior nasal, the extended C1 superior temporal, the extended C1 inferior nasal, and the extended C1 inferior temporal area, 1500 μm concentric apart from the center circle, respectively.

**Table 1 pharmaceuticals-14-01140-t001:** The demographics of patients and control.

	CSC (n = 14)	Normal Control (n = 9)	*p*-Value
Age	59.5 (15.85)	70.4 (7.55)	0.147
logMAR	0.15 (0.21)	−0.004 (0.10)	0.154
Mean refractive error	−0.29 (2.14)	1.22 (1.23)	0.086
Spot size (μm)	4521 (1201)		

Data are presented as mean (standard deviation) and the Mann-Whitney *U* test was conducted.

## Data Availability

Data available within article.

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
