# Peer review of "Choroidal Volume Evaluation after Photodynamic Therapy Using New Optical Coherence Tomography Imaging Algorithm"

_pharmaceuticals, 2021, doi:10.3390/ph14111140_

Round 1
Reviewer 1 Report
The authors describe the evaluation of choroidal volume after PDT using an OCT imaging algorithm. Enclosed are remarks and suggestions for improving the article.
Spelling/grammar overall:
- Abstract line 11: authors write 'thinnest and thickest thicknesses' Minimum and maximum are probably better terms? ‘thickest thickness and thinnest thickness are also mentioned more often in the rest of the article
Introduction:
- Line 30: are these side effects from the pathology or from the PDT treatment?
- Authors introduce half-dose, half-fluence PDT, but do not introduce the standard protocol or the parameters for these adjusted protocols. What is the photosensitizer dose, light dose, light fluence?
Results:
- The results should be divided in subheadings such as: patient population, inter-test repeatability, changes in choroidal volume, case presentations
- Line 68: introduce abbreviation GLD
- Line 97: what is meant with the ~ ?
- Line 66:….. and 0.21 (2.09, range: -5.75 to +2.75 diopters). Add ‘for the control group’.
- What were the specific protocol and parameters (light dose, light fluence, verteporfin dose) for half fluence, half dose and half time?
- If three different treatment groups are included, are there any differential effects between these treatment groups? Separate analyses of these treatment groups would add valuable information to this article.
- To consider: could the data also be presented in for example a bargraph for more rapid and clear visualization and interpretation? The tables could be added as supplementary data.
- Figure 1 and 2: indicate ‘before treatment, + 1 month and + 3 months’ in the figure itself instead of in the legend only. Furthermore, indicate with graphis (arrow, arrowhead) what authors mean in for example f, j, n, g, k, o with ‘shallow subretinal attachment’ and ‘intraretinal hyperreflective materials’
- Refer to figures 3 and 4 in the results section as well, to provide clarity for the different localizations in which measurements were done
- Did authors compare the algorithm method with the manual method and could this be of value to add?
Discussion
- Authors mention limitations of the study, and they also mention the sex bias. Why did they not use only males in the control group?
Reviewer 2 Report
General
This study performs choroidal volume and thickness analysis after PDT for patients with CSC. Several parameters were evaluated before and after one and three months after PDT. The study is rigorous, the paper is sound, while the topics approached are of high interest. The paper can therefore be considered for publication in Pharmaceuticals, with some necessary improvements, as pointed out bellow.
Specific
1) The English has issues, with edits here and there, and grammar errors. Also, some expressions must be checked, for example “poor corporation with taking images” in line 312. Please recheck the entire manuscript in this respect. Please also check from the point of view of the journal’s template.
2) Please provide for readers a few technical parameters of the utilized SS OCT system, including resolution, center wavelength, and penetration depth. Also, please use the common SS notation for the swept source system. Avoid indicating the manufacturer, city, country each time you mention the same system.
3) Please provide more initial refs on OCT when first introducing the technique, including a ref on advantages and characteristics of SS-OCT,
Drexler, W.; Liu, M.; Kumar, A.; Kamali, T.; Unterhuber, A.; Leitgeb, R.A. Optical coherence tomography today: speed, contrast, and multimodality. J Biomed Opt 2014, 19, 071412.
as well as on the maximum resolution of OCT
Cogliati, A., Canavesi, C., Hayes, A., Tankam, P., Duma, V.-F., Santhanam, A., Thompson, K. P., and Rolland, J. P., MEMS-based handheld scanning probe with pre-shaped input signals for distortion-free images in Gabor-Domain Optical Coherence Microscopy, Opt. Express 24(12), 13365-13374 (2016).
Also, please better point out for readers the choice of the OCT imaging technique for eye (in particular retina) research.
4) Please make sure all notations are introduced when first used in the text, for example OCTA, etc. Please check the manuscript in this respect. A table of abbreviations would be useful for readers at the end on the text (before refs), please introduce it.
5) The notation “um” is not appropriate, please correct (especially as in other parts you have “microns”).
6) The Conclusions sections looks way too brief, please make it more comprehensive, pointing out the main findings of the work.
7) At “Institutional Review Board Statement” please provide no. and date of approval.
8) Please do not use “1m” in tables for one month.
9) Please use “p” with italics instead of P at statistics.
Round 2
Reviewer 2 Report
The paper has been substantially revised, according to all the comments and suggestions made. In the opinion of this reviewer, the manuscript can be accepted in the present form for publication.